# Alpha Synuclein: Neurodegeneration and Inflammation

**DOI:** 10.3390/ijms24065914

**Published:** 2023-03-21

**Authors:** Gianluigi Forloni

**Affiliations:** Department of Neuroscience, Istituto di Ricerche Farmacologiche Mario Negri IRCCS, Via Mario Negri 2, 20156 Milano, Italy; gianluigi.forloni@marionegri.it; Tel.: +39-0239014462; Fax: +39-0239001916

**Keywords:** Parkinson’s disease, lewy body dementia, α-synuclein oligomers, oligomeropathies, astrocytes, microglia, enteric nervous system, Disease Modifying Therapy, doxycycline

## Abstract

Alpha-Synuclein (α-Syn) is one of the most important molecules involved in the pathogenesis of Parkinson’s disease and related disorders, synucleinopathies, but also in several other neurodegenerative disorders with a more elusive role. This review analyzes the activities of α-Syn, in different conformational states, monomeric, oligomeric and fibrils, in relation to neuronal dysfunction. The neuronal damage induced by α-Syn in various conformers will be analyzed in relation to its capacity to spread the intracellular aggregation seeds with a prion-like mechanism. In view of the prominent role of inflammation in virtually all neurodegenerative disorders, the activity of α-Syn will also be illustrated considering its influence on glial reactivity. We and others have described the interaction between general inflammation and cerebral dysfunctional activity of α-Syn. Differences in microglia and astrocyte activation have also been observed when in vivo the presence of α-Syn oligomers has been combined with a lasting peripheral inflammatory effect. The reactivity of microglia was amplified, while astrocytes were damaged by the double stimulus, opening new perspectives for the control of inflammation in synucleinopathies. Starting from our studies in experimental models, we extended the perspective to find useful pointers to orient future research and potential therapeutic strategies in neurodegenerative disorders.

## 1. Introduction

Parkinson’s disease (PD) is the second most import neurodegenerative disorder after Alzheimer’s disease (AD), causing progressive movement disability, bradykinesia, tremor, and rigidity, accompanied by non-motor symptoms. The neuropathological hallmarks of PD are the loss of dopaminergic neurons in the substantia nigra pars compacta (SNpc) and the presence of intracellular inclusions termed Lewy bodies (LBs) to honor the neuropathologist who first described these cytoplasmic eosinophilic fibrillary bodies. In the PD brain, LBs and Lewy neurites (LN) were observed in deep mesencephalic nuclei, in SNpc, and the olfactory bulb, spreading to other regions with a prion-like mechanism in patients staged by Braak et al. (2003) [1], and later confirmed for PD patients with early onset and long duration disease, and for Lewy body dementia (LBD). In LBD, there were large amounts of LBs and LN in the cortex, and in multiple system atrophy (MSA) where LBs have also been found in glial cells, mainly oligodendroglia (glial cytoplasmic inclusions, GCI) [2].

LBs were immunocytochemically identified with anti-ubiquitin antibodies [3], but a breakthrough came when molecular genetic investigations demonstrated that autosomal dominant forms of PD were associated with mutations of the gene encoding α-synuclein (α-Syn) [4,5]. Originally sequenced as a precursor of the non-β amyloid component of senile plaque in AD (NACP) [6,7], α-Syn was identified as the main component of LB [8,9], particularly its phosphorylated form at serine residue 129 (p-Syn) [10,11]. Thus, PD and the disorders characterized by LB were collectively named synucleinopathies.

In this first quarter of the century, the biology of α-Syn in physiological conditions and its role in neurodegenerative disorders has been extensively investigated. Alpha-Syn defined natural unfolded protein is localized presynaptically, and its sequence of 140 amino acids can be structurally divided into three domains, the amphipathic N-terminal region with residues 1–60; the central non-Aβ-component (NAC) portion with residues 61–95, which is highly hydrophobic and fibrillogenic, and the acidic C-terminal region with residues 96–140. We have described the distinct functions of these three regions, including a neuroprotective activity restricted to the C-terminal region [12]. In physiological conditions, α-Syn has been described in alpha helical tetramer structures resistant to aggregation [13,14,15]. It has been suggested that tetramers undergo destabilization of their helically folded conformation prior to α-Syn aggregation into abnormal oligomeric and fibrillar assemblies [16,17]. Although the importance of the tetrameric structure of α-Syn in the physiological condition is debated in favor of the more common unstructured monomeric form [18], there is evidence that this structure is sensitive to the presence of various mutations associated with PD [19,20]. Numerous investigations have connected the physiological activity of α-Syn in monomeric forms to synaptic functionality [21,22], although there are still no conclusive results about its physiological function, while the pathological events associated with α-Syn are almost exclusively linked to its polymeric forms. Alpha-Syn can bind to lipid membrane, and adopts highly ordered α-helical conformations on lipid binding; the biophysical properties of this interaction have been extensively characterized and associated with α-Syn aggregation mechanisms [23,24,25].

The identification of α-Syn in cerebrospinal fluid (CSF) [26] and the propagation of LB pathologies from transplant to host cerebral tissue in humans [27,28] substantially changed the view on synucleinopathy etiology, and although the accumulation of α-Syn is intracellular, they became similar to the other central amyloidoses. Thus α-Syn oligomers (α-SynO) are responsible for pathological spreading and neuronal dysfunction through two independent mechanisms, as shown in prion diseases, where propagation of prion is distinct from the neurotoxic activity [29]. Furthermore, as discussed below, α-SynO are also important in the neuroinflammation associated with selective neurodegeneration in the early phase of PD and related disorders [30]. In various experimental conditions, it has been reported that α-SynO in combination with neurotoxic activity can induce glial activation [31]. On the other hand, general inflammation can influence the biological activity of α-SynO with positive or detrimental consequences for the neurotoxic processes [32,33]. The crosstalk between α-Syn and the immune system involving peripheral and central compartments is complex and fundamental in the pathogenesis of synucleinopathies [34,35,36] (Figure 1).

## 2. Alpha-Synuclein Oligomers

The self-assembling capacity of amyloidogenic peptides follows the nucleation mechanism proposed by Lansbury’s group thirty years ago [37]. The initial aggregation, seeding, involved a few molecules that change conformation to β sheet, from soluble oligomers to protofibrillar and fibrillar insoluble structures. Besides the common tendency to shift from random coil to β-sheet structure, the size and characteristics of monomeric proteins in neurodegenerative diseases vary widely, and a “one-dimensional crystallization” model with minimal variations [37,38] is still quite capable of describing the chemico-physical mechanism of the formation of insoluble aggregates.

The seeding mechanism responsible for protein deposition can be seen in diverse cellular compartments, with no substantial differences, and numerous factors including pH, ionic strength, temperature, mutations in amino acid sequence, metals and concentration can affect the amyloidogenesis. The pathological form of α-Syn assembles in prion-like polymeric structures by inducing the conformational conversion of native α-Syn. The process of α-Syn polymerization through oligomeric to insoluble fibrillar structures is complex, and the intermediate conformations are heterogeneous.

The recent development of cryo-EM and image processing techniques has permitted ultrastructural analysis with a resolution close to the atomic scale. These innovative techniques have been employed in various experimental conditions to analyze the ultrastructural differences in α-Syn fibrils, revealing interesting fundamental aspects of the polymorphic capacity of α-Syn aggregates. Cryo-EM analysis of recombinant α-Syn fibrils in cell-free conditions confirms the structural heterogeneity of wild-type and mutated sequences [39,40,41,42]. The analysis of fibrils from protein misfolding cyclic amplification starting from PD or MSA tissue homogenate with recombinant α-Syn showed differences from those generated only from recombinant seeds [43]. A recent cryo-EM study on α-Syn filaments from the brains of individuals with PD, PD with dementia and DLB showed them to be a single protofilament (Lewy fold), markedly different from the protofilaments of MSA [44]. Moreover, the fibrils purified from the brain present different structures from those generated in vitro using recombinant α-Syn seeded with previously extracted fibrils [44,45].

These findings confirm the structural polymorphism in relation to disease PD/LBD vs. MSA, but it is important, on the other hand, to note that multiple structures found by assembling recombinant α-Syn in vitro did not show similar heterogeneity in the analysis of brain extracted filaments. More results are needed to clarify the extent to which the structure of the fibrils formed in vitro reproduces the structural characteristics of in vivo-formed fibrils, but it appears that the nature of α-Syn filaments is less variable than expected, and similar conclusions are shared by the cryo-EM analysis of other amyloid structures [46,47,48].

The central NAC portion is the highly amyloidogenic part of α-Syn, driving the conformational change from random coil to β-sheet structure to fibrils [49]. Oligomeric structures are heterogeneous and metastable; their progression to mature fibrils is not inevitable, and alternative pathways to oligomer conformers have been described [50]. Alpha-Syn undergoes spontaneous assembly, and its aggregation capacity is influenced by mutations associated with familial PD [51,52], leading to the formation of annular protofibrils resembling a class of pore-forming bacterial toxin. Inappropriate membrane permeabilization, therefore, could be responsible for cell dysfunction and cell death [53,54] (Figure 2).

The heterogeneity of α-SynOs, influencing their toxicity, arises during their assembly. There is compelling evidence of their pathogenic role in synucleinopathies compared with monomers or fibrils that can adopt different morphologies such as spherical, chain-like, annular (pore-like structure), and tubular [17,55]. In addition to sequence mutations, α-Syn aggregation is also influenced by various post-translational modifications including phosphorylation, ubiquitination, nitration, and oxidation, which have been identified in experimental models as well as brains from PD patients. The phosphorylation at codon 129 has been involved in PD pathogenesis, but recent investigations show that this phosphorylation occurs when the aggregation is well established, and even inhibits the assembling capacity of α-Syn in the initial aggregation steps [56]. Acetylation at the N-terminal does not influence, or in some circumstances reduces, the aggregation capacity of the α-Syn full-length sequence [57,58,59].

The studies on ubiquitination are contradictory: in some cases, the self-assembling capacity of α-Syn was reduced, and in others, enhanced [58]. It has also been shown that ubiquitination may favor the α-SynO removal by proteasome [60]. More consistent are the results on the effect of glycation, which enhances the aggregation and toxicity in vitro [61] and promotes oligomerization [62]. SUMOrylation is another important post-translational modification of α-Syn that affects its accumulation, aggregation and toxicity [63].

We used the term *oligomeropathies* to stress the fundamental role of the oligomeric forms in the pathogenesis of virtually all protein misfolding neurodegenerative disorders; although the proteins involved in the diseases are different, the propensity to aggregate is what substantially activates a common mechanism of polymerization [38,64,65]. The nucleation of α-Syn in a structure progressively enriched with the β-sheet component is based on the initial self-assembly of a few monomers that create the oligomeric structures, heterogeneous by definition [55,66].

The affinity for membrane structures has been shown for AβOs and other oligomeric species, and α-SynOs can affect the membrane integrity by insertion in lipid bilayers [67]. However, this interaction accounts only partially for the α-SynOs’ neurotoxic effects [66]. The α-SynO oligomerization process is preferentially located on the membrane surface, and the mitochondrial membranes are more vulnerable to permeabilization [23,68,69]. This can be easily associated with the particular vulnerability of mitochondria in PD pathogenesis. Thus, α-SynOs became a potential therapeutic target in synucleinopathies [70,71].

### 2.1. Oligomers and Transmission of LB Pathology

In formalin-fixed paraffin-embedded brain sections from patients with PD, a novel technique to detect α-SynOs has been used to examine the distribution of α-SynOs [72]. The authors observed a difference between the distribution of α-SynOs and Lewy pathologies in PD. Alpha-SynOs were more widely distributed than Lewy pathologies, suggesting that α-SynOs may be found throughout the brain sooner in the disease course than can be detected with immunohistochemistry for Lewy pathologies.

Given the toxicity of α-SynOs, clinicopathological studies focusing on α-SynOs may give information on PD pathogenesis. These results suggest that α-SynOs may be widely distributed in PD early in the disease, and they may contribute to cognitive impairment in PD [73]. The presence α-SynOs was confirmed in the hippocampus of patients with cognitive impairment.

This important observation found direct support from our results in transgenic mice carrying the A53T human mutated sequence of α-Syn. The mutation is associated with early-onset PD, and in transgenic mice it induces cognitive decline and an accumulation of α-SynO oligomers in the hippocampus and cortex [74]. This evidence, apparently conflicting with a PD general model where the cognitive decline is secondary to motor impairment, is consistent with recent findings supporting the pathogenic role of α-SynO oligomers in PD [75]. This study used different forms of α-Syn aggregates assembled in vitro or purified from PD brain material and showed that soluble non-fibrillar α-Syn aggregates shorter than 200 nm were highly toxic species, causing lipid membrane permeabilization and inflammation. Furthermore, the hippocampus contained a higher proportion of smaller aggregates than the visual association cortex, and these hippocampal aggregates caused an increased inflammatory response [75].

It has been proposed that PD pathology can spread with a prion-like mechanism through the olfactory and vagal systems to the substantia nigra. The presence of α-Syn deposits in the olfactory bulb and anterior olfactory nucleus at Braak stage I supports this [76]. Similar evidence was reported in LBD [77]. In experimental studies, the early appearance of α-Syn deposits in the olfactory bulb has been described in transgenic mice PD models [78,79,80]. In these models, the neuropathological alterations were associated with hyposmia that appeared before the other behavioral alterations [81]. Injections of mouse or human α-Syn fibrils into the olfactory bulb of wild-type mice recruited endogenous α-Syn into pathological aggregates that spread transneuronally to over 40 other brain regions and subregions over the course of 12 months [82,83]. Inoculation of α-Syn preformed fibrils (PFF) in the olfactory bulb of non-human primates (NHP) induced a spreading of α-Syn pathology in several brain regions including SNpc and the dorsal nucleus of vagus [84].

As mentioned above, the intracellular localization of LBs and LNs in synucleinopathies implies that their formation is the consequence of a cell-to-cell passage of α-Syn. Thus, neuronal dysfunction resulting from exposure to neurotoxic oligomeric species is combined with the seeding mechanism activated by fibrils and oligomers [85,86]. Danzer et al. (2007) [87], using different protocols, prepared various α-SynO conformers and compared their biological activities in terms of cell death, cytosol calcium levels and intracellular α-Syn accumulation. In SHSY-5Y cells and primary cultures, smaller, annular structures affected calcium influx and cell death, while larger structures exerted no neurotoxic activity but promoted intracellular α-Syn aggregation, as described by Lee et al. (2010) [88]. After α-SynO exposure, the intracellular formation of large aggregates was associated with a reduction in cytoplasmic α-Syn staining, while exposure to the fibrillar form of α-Syn showed no detectable seeding effect [89]. The effect was dose- and time-dependent, and the authors demonstrated that small oligomeric structures can be converted to larger formations, activating a seeding mechanism [88].

In cultured cells overexpressing α-Syn, intracellular aggregation has also been triggered by the introduction of recombinant α-Syn fibrils incorporated in liposomes [90]. Alpha-Syn fibrils recruited endogenous soluble α-Syn proteins and converted them to insoluble, hyperphosphorylated, and ubiquitinated pathological species [90]. The exposure to α-Syn short amyloid fibrils induced intracellular aggregation in SHSY-5Y cells [89]. Alpha-SynO is secreted from neuronal cells by nonconventional exocytosis involving extracellular exosomes [91] and internalized through an active process associated or not with exosomes through an endocytic pathway [86,92,93,94]. The aggregated forms of α-Syn are associated with exosomes externally on the membrane surface or inserted within the membrane; however, internal accumulation also has been observed [86,95,96] (Figure 2).

These findings indicate that the spreading of α-Syn pathology from a cell-to-cell activating seeding intracellular mechanism is mainly associated with α-Syn in the oligomer conformation, which might be incorporated through different biological mechanisms and secreted from neuronal cells (Figure 2). In several experimental conditions, the intracellular cellular aggregation of α-Syn by exposure to PFF has also been shown [97,98,99].

### 2.2. Oligomer Neurotoxicity

The possibility that α-Syn might exert direct neurotoxic activity was immediately considered when the combination of genetic and neuropathological findings suggested a central role of this protein in the PD pathogenesis [100]. Since the initial studies in 2000, the possibility has been suggested that the oligomeric form of α-Syn is the main cause of neuronal dysfunction [101]. Using NAC peptide associated with a TAT sequence to vehicle the peptide inside the cells, we demonstrated selective dopaminergic toxicity in primary mesencephalic cultures and by intranigral injections in vivo [102]. Similarly to the previous investigations with Aβ peptides, α-SynO conformers were investigated in various conditions [103,104,105,106] and the possibility that different species had distinct biological activities was tested [87]. As mentioned before, the intracellular accumulation of α-Syn in pathological conditions made it harder to immediately understand the sequence of pathological events associated with α-Syn going from monomeric to insoluble fibrillar structures. The recent demonstrations that toxic oligomers can be released from fibrillar structures in cell lines and human iPSC model [66,107] and that oligomers can induce axonal damage and direct mitochondrial alterations [108] are important achievements that help to clarify the pathological process in synucleinopathies.

Recently, it has been shown that the activity of tyrosine hydroxylase, an enzyme specifically expressed in catecholamine-containing neurons, can induce the hydroxylation of Tyr 136 in the α-Syn sequence. This DOPAnization promoted the α-Syn oligomerization, which may play a pathogenic role in dopaminergic neurodegeneration [109]. The Y136-DOPA-Syn signal was detected at the striatonigral axon terminals of aged A53T Tg mice and patients with PD or MSA but not in controls. This suggests that DOPAnization of α-Syn may occur during the disease progression rather than initiating the pathological process, but in any case, it might contribute to the selective loss of dopaminergic neurons in synucleinopathies [109].

The neuronal dysfunction induced by exogenously applied α-SynOs has been investigated in various experimental conditions. We have developed a model with intracerebroventricular (ICV) injection of α-SynOs followed by behavioral, biochemical and histopathological examinations. In similar conditions, Martin et al. (2011) [110] described cognitive deficits and LTP inhibition induced by α-SynOs involving calcineurin. According to Fortuna et al. (2017) [111], a single ICV injection of α-SynOs recapitulated some of the PD pathological features, including dopaminergic cell loss and olfactory dysfunction together with motor behavior alterations. The basic pathological PD features have also been described at the experimental level with intranigral injections of α-SynOs [112], though in some conditions, α-Syn fibrils too can induce neurotoxicity [113,114,115,116].

In an experimental model developed to study the biological activities of Aβ oligomers [117,118], we compared the effects of α-Syn in monomeric, oligomeric and fibrillar forms injected ICV in wild-type mice, followed by behavioral and histopathological examinations. In contrast with the monomeric and fibrillar forms, which were ineffective, α-SynOs induced cognitive decline, seen in the object recognition test (ORT); the effect was reversible and specifically inhibited by anti- α-Syn antibody [31] and was associated with hippocampal glial activation, as observed in PD brain [75].

The neurotoxic activity of oligomers is the consequence of various interactions with cell function. Together with non-specific interferences with cell membrane that can alter its chemico-physical structures, affecting its microfluidity [119,120,121,122], two more specific mechanisms have been observed [123]. Oligomers, including α-SynOs, can directly interact with entities activating specific intracellular pathways in different conditions, although the functional consequences of this receptor interaction were not always confirmed [74,118]. Finally, several oligomeric structures formed membrane pores with ion channel function [17,87,124,125,126].

Figure 2 summarizes the possible neurotoxic mechanisms associated with exposure to α-SynOs. In neurons, α-SynOs induce perturbation of cellular and mitochondrial membranes, altering their structures and functions [71,105,127,128,129,130]. Specific receptor interactions have been proposed to explain α-synO toxicity, including prion protein [131,132], β spectrin [133,134], RLP1 receptor [135] and purinergic P2X7R receptor [136,137]. Interaction with the GluN2A NMDA receptor subunit caused synaptic dysfunction in striatum [138]. As discussed below, an essential contribution to α-synO toxicity comes from glial activation and the production of inflammatory factors affecting neuronal function (Figure 1). In this pathological scenario, the activation of TLR-2 by α-SynOs is particularly important.

## 3. Alpha-Synuclein and Inflammation

### 3.1. Peripheral and Central Inflammation

Inflammation influences the pathogenesis of virtually all neurodegenerative disorders; distinct inflammatory pathways are activated, and the timing can be different, but local or/and general inflammation is invariably involved, starting from the early phase of diseases [139]. In PD and the other synucleinopathies, dysfunction of the central and peripheral immune systems significantly participates in the development of the diseases [140].

In relation to inflammation, the increasing importance of two-way communication between the gut and the brain in neurodegenerative disorders has been supported by consistent results and numerous theoretical suggestions that await proof from experimental data [141]. However, in PD, the peripheral component of the disease is well established. PD can be considered a multi-system disorder with neuroinflammation and immune dysfunction implicated in the development of various non-motor symptoms, such as sleep and gastrointestinal dysfunction, which can precede the disease diagnosis by decades [142,143,144]. Disturbances in the gut–brain axis have been found to be associated with PD; neuropathological changes in the enteric nervous system (ENS) and significant alterations of the gut microbiota suggest an important role of these systems [145,146]. Alpha-Syn appears to have a fundamental role in the deleterious crosstalk between neuronal and immune system dysfunction in the early stages of PD pathology involving proinflammatory factors and the gut microbiota.

In the early staging of PD, α-Syn pathologies in the olfactory nucleus and vagal nerve were identified [1,147], then α-Syn inclusions in the myenteric and submucosal plexus of the ENS of postmortem PD patients [148]. This proves that α-Syn pathogenesis might originate in the gut, then propagate to the CNS.

The possibility that PD may begin with α-Syn aggregation in the ENS and then propagate to the CNS via the vagus nerve has been suggested by numerous findings [144]. In a first experiment, brain lysates from PD patients containing different types of α-Syn aggregates or synthetic PFFs were injected into the gastrointestinal wall of rats, and the transfer of aggregated α-Syn from the gut to the brain along the vagal nerve was observed [149].

However, direct induction of α-Syn pathology in the ENS by α-Syn viral overexpression or PFF injections in rodents and non-human primates (NHP) induced local pathology that did not spread outside the ENS. Rats and NHP received targeted enteric injections of PFF or adeno-associated virus overexpressing the A53T α-Syn mutated sequence associated with PD. Histopathological examination at different times after the injections showed α-Syn transient pathology in the brainstem (dorsal motor nucleus of the vagus and locus coeruleus) one month later; however, no pathology was observed at later times. Similarly, the examination of brains from injected NHP revealed no pathology despite the robust α-Syn accumulation throughout the ENS that persisted for the entire study (12 months) [150]. The authors, therefore, suggested that sustained spread of α-Syn pathology from the periphery to the CNS and subsequent propagation is rare, and that the presence of enteric α-Syn pathology and dysfunction may be an epiphenomenon [150]. This was supported by an investigation in peripheral autoptic material from PD and LBD subjects compared to age-matched controls. The staining of p-Syn, a sensitive marker of α-Syn pathologies, was seen in the vagus in 46% of LBD subjects and 89% of PD subjects, and 17% and 81%, respectively, in the stomach; no p-Syn was found in controls. The authors concluded that the involvement of peripherical organs did not distinguish all PD/LBD subjects from controls, and consequently, synucleinopathies must first be in the brain [151]. A more rational interpretation is that in the majority of DLB cases and less than 20% of PD cases, α-Syn pathology propagation is exclusively in the CNS, but although the peripheral involvement is partial, we could not rule out a peripheral start of the synucleinopathies, coexisting with cases originating inside the brain [152].

In another NHP model, either intrastriatal or enteric injection of LB extracts from PD patients induced nigrostriatal lesions and ENS pathology [153]. These findings indicated the two-way long-distance propagation of α-Syn pathology between the CNS and the ENS in PD, varying between patients and disease subtypes. However, α-Syn pathological lesions were not found in the vagal nerve, excluding this route of transmission of PD pathology.

These results suggest a possible systemic mechanism in which the general circulation acts as a route for long-distance two-way transmission of endogenous α-Syn between the enteric system and the CNS [153]. Brain extract from PD subjects was injected in this case into either the brain or ENS, while the previous investigation used recombinant α-Syn or adenovirus to induce the mutated sequence of α-Syn. We cannot rule out that other components in the cerebral material might favor the transmission of the pathology. However, in rodent models, α-Syn PFFs also propagated α-Syn pathology from the periphery to the brain. Kim et al. (2019) [154] injected α-Syn PFFs into the muscularis layer of the duodenum and pylorus in mice, showing gut-to-brain transmission of α-Syn pathology. Van Den Berge et al. (2019) [155], injecting PFFs into the wall of the duodenum in transgenic rats overexpressing human α-Syn, reported wide distribution of α-Syn pathology, including the brainstem.

It is important to stress that in these experimental studies, α-Syn transmission and deposition were constantly associated with inflammation induced by α-Syn, and in a vicious circle, inflammation promoted the propagation of α-Syn pathology [156,157,158], and intestinal inflammation increased the cerebral α-Syn accumulation [159].

### 3.2. Inflammation and α-Syn oligomers

The glial activation induced by ICV injection of α-SynOs was determined biochemically and immunohistopathologically, measuring the expression of IBA-1 and GFAP as markers of astrocytes and microglia, respectively, in hippocampus and cortex [31,36] (Figure 1).

In neurodegenerative disorders, the activation of TLR-2 and TLR-4 influenced the pathogenic process negatively [160,161,162,163,164,165]. We distinguished a role of TLR-4 in the AβOs and TLR-2 in the toxic effect of α-SynOs by measuring cognitive decline found similar results when glial activation was considered. The TLR-2 antagonist inhibited the glial activation induced by α-SynOs, while the TLR-4 antagonist or TLR-4 silencing abolished the glial reaction induced by AβO [17,31].

The NLRP3 inflammasome, an important component of the innate immune system, is a multimeric protein complex that assembles in response to homeostasis-altering molecular patterns (HAMPs) and certain other cellular insults [166]. The stimulation of TLRs is part of the activation of the nucleotide-binding oligomerization domain-leucine-rich repeat-pyrin domain-containing 3 (NLRP-3) inflammasome [167]. Alpha-Syn aggregates induced the secretion of proinflammatory cytokines such as IL1β and IL18 through the activation of inflammasomes, particularly the NLRP-3 [168,169]. Systemic NLRP3 inflammasome expression and the increase in IL-1β levels in the serum negatively correlate with PD progression [170].

Oral treatment with an NLRP3 inhibitor, MCC950, in wild-type mice injected with fibrillar α-Syn reduced the inflammatory factors but also the accumulation of α-Syn aggregates and dopaminergic damage [171].

Microglia are the primary innate immune cells in the brain, with an important role in maintaining cerebral homeostasis by detecting changes in their surroundings, removing cellular debris, and providing neurotrophins [172]. However, when microglia are activated by pattern recognition receptors (PRRs), they can trigger an inflammatory response that can lead to chronic neuroinflammation and neuronal damage. This chronic neuroinflammatory response is observed in patients with synucleinopathies, where microglia activation occurs in all brain regions accumulating α-Syn. Alpha-Syn can induce excessive microglial activation and inflammatory activity, which can contribute to the progression of neurodegenerative diseases [173] (Figure 1). The accumulation of α-Syn induces the secretion of IL-1β by microglia cells through activation of the NLRP3 inflammasome. Inhibition of this inflammasome has been found to prevent α-Syn pathology and dopaminergic neurodegeneration in mouse models of PD [171].

The manipulation of inflammasome at various levels improved the overall clearance of α-SynOs [173,174]. This effect was initially restricted to the activation induced by α-Syn in fibrillar form [175] but the recent results by Scheiblich et al. (2021) [163] extended the capacity to activate microglial inflammasome at the α-SynOs. The inflammasome activation by α-SynOs in microglial cells has been associated with inhibition of an autophagic mechanism and increased TNF-α release [176,177,178]. The activation of autophagy, on the other hand, affects the accumulation of α-Syn [179] overexpression of the Bcl2-associated athanogene (BAG)3 promoted autophagy and reduced the activation of the NLRP3 inflammasome induced by LPS regulating autophagy [180].

Gut inflammation may be an important driver of the systemic immune response in PD. Activation of NLRP3 inflammasome in PD has also been related to peripheral inflammation, and the microbiota–gut–brain axis has been implicated via enteric bacterial regulation of this inflammasome [181]. The levels of α-TNF, γ-interferon, IL-6 and IL-1β negatively correlated with disease duration [182].

### 3.3. Role of Astrocytes

The influence of general inflammation on the biological activity of α-SynOs in CNS was directly investigated in an experimental model in wild-type mice where a lasting level of inflammation was induced by a single peripheral injection of lipopolysaccharide (LPS). In comparison with control one month later, the effects of ICV injection of a sub-active dose of α-SynOs were tested by histological and behavioral examinations. The combination of LPS and a sub-active dose of α-SynOs induced cognitive decline and amplified microglial activation in the hippocampal and cortical regions, whereas astroglial reactivity was reduced with the double-hit compared to the single treatment [36]. Different behavior of astrocytes and microglia was also observed in α-Syn A53T transgenic mice injected with LPS, where microglial activation was exacerbated, while astrocyte activation was abolished.

These findings clearly illustrate the relationship between systemic inflammation and central action of α-SynOs. In our model, LPS peripheral treatment induced glial reactivity per se; the synergic effect can be restricted to the CNS. As reported by Garcia-Revilla et al. (2022) [183] in several experimental models including rotenone and 6OHDA injections, the influence of general inflammation on central neurotoxic activity is well established. The different reactivity of astrocytes and microglia is important evidence that needs to be considered in the PD pathogenesis; the astrocyte dysfunction might contribute to the neurodegenerative process in synucleinopathies [184,185].

As shown by Kuter et al. (2018) [186], prolonged astrocyte dysfunction combined with microglia activation accelerated neuronal degeneration. iPSC-derived astrocytes from PD patients with G2019S-LRRK2 mutation showed morphological atrophy and mitochondrial malfunction compared to those from age-matched control donors [187].

Brandebura et al. (2023) [188] have extensively analyzed the role of astrocytes in neurodegeneration. They underlined the changes in astrocytes at different stages of disease progression and the heterogeneity of their responses in different brain regions, distinct from any single neurodegenerative disorder. The contribution of astrocytes to neurodegeneration could be summarized in two directions: their dysfunction, which alters the physiological homeostasis indispensable for correct cerebral function, and their reactivity, which induces the production of harmful factors. Thus, alterations to astrocytes encompass both toxic gain of function and impairment of essential function. Our data fit well in this scenario, considering the astrocyte dysfunction induced by the combination of general inflammation and α-SynOs as the first part of the pathogenic mechanism in PD.

## 4. Therapeutic Perspectives

### 4.1. Synuclein Aggregation as a Pharmacological Target

Treatments to control the central and peripheral symptoms of PD are efficient and are divided to follow the progression of the disease. The numerous pharmacological and non-pharmacological tools achieve an adequate level of care especially in the early phase of the disease [189]. However, similarly for other neurodegenerative disorders, we still lack a disease-modifying drug or treatment to interfere with the pathogenesis, arresting or slowing its progression. The identification of α-Syn as the main component of LB led to experimental models being developed to demonstrate the neurotoxic activity of α-Syn in a mutated or wild-type sequence [102,190,191], and cerebral accumulation of α-Syn was immediately considered as a pharmacological target in PD and other synucleinopathies. Direct anti-amyloidogenic activity against α-Syn aggregation or modulation through the HSP70/90 pathway was initially proposed as a therapeutic approach [192,193,194,195]. Small molecules, peptides, and peptidomimetics directly targeting α-Syn and the aggregation pathways were investigated [196]. Low-molecular-weight compounds with different chemical structures have been screened in vitro, measuring their anti-aggregation activity against α-Syn’s self-assembling capacity. In the study by Masuda et al. (2006) [197], 79 compounds belonging to 12 different chemical classes inhibited α-Syn filament formation in vitro: these included polyphenols, phenothiazines, porphyrins, polyene macrolides, and Congo red and its derivatives. As expected, several of these were also able to inhibit the self-assembling aggregation of tau and β amyloid.

Other high-throughput screening methods followed to identify anti-Syn aggregation compounds [198,199], including some recent ones where known [200] or innovative methods [201,202] were applied. Numerous potential candidates to interfere with synucleinopathy pathogenesis were identified, but only a few have reached the clinical stage [203]. These include anle138b, a lead compound from a systematic high-throughput screening test combined with medicinal chemistry optimization, which showed anti-oligomerization activity against prion and α-Syn aggregates [204]. The intimate mechanism of interaction of anle138b with α-Syn fibrils has been investigated with cryo-EM, revealing stable polar interactions between anle138b and backbone moieties inside the tubular cavity of the fibrils [205]. In different PD mouse models, anle138b strongly inhibited oligomer accumulation, neuronal degeneration, and disease progression in vivo [204,206,207] and motor deficits in MSA models [208,209,210]. These findings, the optimal blood–brain barrier (BBB) passage and the oral administration strongly supported clinical investigations to test the efficacy of anle138b in PD. Two phase 1 randomized trials are evaluating the safety, tolerability and pharmacokinetics of anle13, and recruitment of the subjects has been completed in both trials. The first reported favorable safety, tolerability and pharmacokinetic parameters [211], and the second phase 1 study will establish the best drug dosage.

Squalamine is another interesting molecule that affects α-Syn aggregation, displacing α-Syn from lipid membranes, altering the first steps in its aggregation [212,213,214]. In an experimental PD model with A53T α-Syn transgenic mice, oral squalamine restored gut motility [215]. A derivative of squalamine, ENT-01, gave positive results on constipation in PD patients [216], and the results of a phase 1 clinical trial to evaluate the efficacy in PD dementia are under analysis [203]. Another aminosterol, trodusquemine, has been tested at the preclinical level to investigate its mechanism of action in inhibiting α-Syn aggregation and its efficacy in animal models [217,218] In artificial membrane, the drug is homogeneously distributed, influencing the lipid raft organization in neuronal membranes and interfering with the start of α-SynO formation [219].

The inhibition of α-Syn aggregation by peptidergic compounds is based on the synthesis of amino acid sequences targeted on the NAC core, from residues 68 to 78 of α-Syn, which bind seed fibrils to prevent their growth and elongation [220,221]. A similar approach was proposed in the past with *β*-sheet breakers in AD, targeting the β amyloid sequence, with interesting results in experimental models [222]. These were replicated in a prion disease model [223] but have not been translated in clinical studies.

Several other peptidergic sequences inhibited α-Syn aggregation and antagonized cell toxicity [224,225,226]. However, moving from the promising peptide sequences to an efficacious therapeutic tool is complex, and several biological obstacles must be removed, including the actual bio-availability and the BBB passage.

### 4.2. Immunotherapy

The immunotherapy in the AD model in the pioneer study by Schenk et al. (1999) [227], used active immunization with an Aβ sequence that reduced cerebral amyloid plaques and antagonized the cognitive behavior deficits [228].

After two decades and numerous failures, immunotherapy is now starting to be considered in AD [229]. Lecanemab, a new-generation anti-Aβ antibody, has been recently approved by the FDA for early AD treatment. Active immunization in synucleinopathies was proposed in animal models a few years after the initial experiments in an AD model [230], and passive immunization was tested later [231]. In both cases, the treatment relieved the histopathological and behavioral consequences induced by the human transgene.

Active immunization with AFFITOPE PD1A (PD01A), a new-generation immunogen, has been tested in various PD models, with positive results in terms of reduction of α-Syn deposition, protection of dopaminergic neurons and improvement of motor behavior alterations [232]. These findings have led to several phase 1 clinical trials with PD01A to identify the dosage regimen for an adequate immunological reaction and to minimize side effects in PD [233] and MSA patients [234]. Another molecule, PD03A, designed with the same technology, has also given positive results, confirmed in PD [235]. Though PD03A vaccination generated α-Syn-reactive IgG in 58% of treated patients, the titers were lower than with PD01A, where the responder rate was 89%. Clinical trials to assess the efficacy of PD01A or PD03A in synucleinopathies have not yet started.

Another synthetic peptide-based vaccine, UB-312, which targets a 12-amino-acid sequence in the C terminus of α-Syn, has been investigated in a phase 1 clinical trial with positive results [236]. The immunization with UB-312 in a Thy1SNCA/15 mouse model of PD prevented functional deficits and both central and peripheral pathology, including a reduction of α-SynO in several brain regions [237,238].

A passive immunization approach using anti-α-Syn antibodies to prevent aggregation and cell-to-cell spread of the disease has also been developed for clinical investigation. Prasinezumab is a humanized antibody that recognizes the C-terminal domain of α-Syn, based on the murine antibody 9 × 10^4^ amply characterized in preclinical studies [231] In a phase 1 trial, prasinezumab was well tolerated and significantly reduced serum α-Syn [239]. Another trial replicated safety results but with no reduction of α-Syn in CSF [240]. The efficacy of prasinezumab was recently tested in a clinical trial in PD subjects, but the drug had no substantial effect on global or imaging measures of PD progression compared with placebo, though there was some reaction with i.v. infusion [241].

With a procedure similar to that for the purified anti-Aβ antibody aducanumab to treat AD [242], cimpanemab was obtained from isolated B-cell lines from healthy individuals. using direct ELISA [231]. The antibody recognized the N-terminal domain of the α-Syn sequence, the epitope within the initial 15 amino acids. Preclinical investigations with cimpanemab indicated some protective effects against α-Syn PFFs injected in α-Syn A53T transgenic mice [243]. A phase 1, single ascending-dose study showed that cimpanemab was well tolerated, although the CSF drug level was less than 1% of the level in plasma [244]. However, a recent phase 2 trial missed its primary and secondary endpoints, did not achieve proof of concept, and the drug was, therefore, dropped from further development [245].

MEDI1341 is another interesting anti-α-Syn antibody recently characterized in preclinical investigations, which showed a picomolar affinity for both monomeric and aggregated forms of α-Syn; in vitro and in vivo models MED1341 reduced α-Syn propagation [246]. A couple of phase 1 trials with MEDI1341 are ongoing in controls and PD subjects.

Several other anti-α-Syn antibodies are in preclinical characterization [247] but, as discussed by Jensen et al. (2023) [248], immunotherapy in synucleinopathies is not a simple issue, and before we see real positive results in clinical trials, several aspects need to be addressed. As discussed in relation to AD [249] there are various reasons for failure, including not only the complexity of the disease only partially known, but also the correct identification of the target, the timing of treatment, selection of patients, appropriate biomarkers, and correct trial design. These reasons are common to any other treatments in neurodegenerative disorders, but immunotherapy is also limited by the scarce passage of the antibodies into the brain (often, the levels of antibodies in CSF are less than 1% of those in serum) and their stability [250]. In addition, an obvious difficulty is solving a complex disease such as a neurodegenerative disorder with a single medication; combination therapy focusing on different biological targets has more chances of success.

### 4.3. Anti-Inflammatory Treatments

The important role of inflammation in synucleinopathies offers the opportunity to consider the control of immunoreactivity and glial activation as pharmacological targets and to develop combined therapeutic strategies focused on α-SynOs [251].

As mentioned before, ICV α-SynOs induced cognitive decline with glial activation, both effects being antagonized by TLR-2 antagonist [31]. A key role has been suggested for TLR-2 in synucleinopathies, in the CNS and periphery associated with various biological activities [173,252,253,254]. This includes a role of α-SynOs as endogenous agonist of TLR-2 on microglial cells [255] with high affinity [256]. The inhibition of autophagy by TLR-2 activation through regulation of the AKT/mTOR pathway is another mechanism acting on α-SynO accumulation, reducing the clearance of α-SynOs [257].

The different mechanisms involving α-SynOs influenced by TLR-2 include microglia and astrocytic activation, neuronal autophagy, and α-Syn transmission from neuron to neuron or from neurons to glial cells [258]. In iPSCs, the activation of TLR-2 acutely impaired the autophagy lysosomal pathway and markedly potentiated α-Syn pathology seeded with α-Syn PFF, while a novel small molecule antagonist of TLR2 (NPT1220–321) strongly reduced the effect [259]. Kouli et al. (2019) [260] found that in synucleinopathies, the effect of TLR-2 restricted to the CNS was associated with an important peripheral role of TLR-4. Xia et al. (2021) [261] reported that plasma exosomes from PD patients injected intrastriatally induced microglial activation and the propagation of exosomal α-syn from microglia to neurons; this effect appeared to be mediated by TLR-2 in microglia. These findings are encouraging the immunotherapeutic approach to PD with molecules that interfere with TLR-2.

The ICV injection of α-SynOs induced parallel neuronal dysfunction and glial activation in hippocampus, whereas systemic treatment with classical anti-inflammatory drugs such as indomethacin or ibuprofen completely abolished the glial activation, as expected, but also antagonized the memory deficits induced by oligomers injected in wild-type mice [31]. This is consistent with the fact that the well-known inflammatory agent LPS per se induces memory impairment [262]. Thus, glial activation induced by α-SynOs seemed essential for the detrimental effect on cognitive performance, and could be involved with TLR-2 at the neuronal but also the glial level, as seen above [263]. Our model simplifies the pathogenic mechanism in PD and LBD and the role of α-SynOs. However, a recent histopathological investigation in humans by Sekiya et al. (2022) [73] showed accumulation of α-SynOs in the hippocampus of PD subjects with memory impairment but also an inflammatory reaction.

We recently investigated the potential therapeutic properties of doxycycline (Doxy) in human α-syn A53T transgenic mice. Doxy is a second-generation semisynthetic tetracycline with several other pharmacological properties besides its anti-microbic capacity. These activities have led to the drug being repurposed to treat several other pathological conditions [262,263,264,265].

The combination of anti-inflammatory and anti-amyloidogenic activity turned out to counteract behavioral, biochemical and histopathological alterations in α-syn A53T transgenic mice [266]. The mice treated with Doxy were rescued from cognitive and daily activity deficits, and although no changes were observed in α-Syn deposits in the SNpc, the drug substantially reduced hippocampal and cortical α-SynO accumulation and glial activation and exerted a neuroprotective effect, overcoming LTP alterations. It is interesting that in an APP/PS1 transgenic mouse model of AD as well, Doxy’s positive effect on cognitive behavior was not immediately associated with a reduction in Aβ deposits when the drug was administered acutely or chronically for a relatively short time; the Aβ plaques were reduced only when treatment was prolonged to three months [267]. Apparently, α-SynOs are Doxy’s main target, and the action on fibrillar deposits becomes evident only when the lack of fuel has slowly reduced their size.

Our studies in different pathological contexts indicate that Doxy can directly interact with oligomers, interfering with fibrillogenesis [268]. González-Lizárraga et al. (2017) [269] showed that Doxy can modify the structure of α-SynOs into off-pathway, high-molecular-weight species; this conformer did not progress to fibrils and was not neurotoxic. This important result might explain Doxy‘s efficacy in synucleinopathies [270,271].

In a recent study by Dos Santos Pereira et al. (2022) [272] Doxy affected L-DOPA-induced dyskinesia (LID) in mice with dopaminergic impairment induced by intrastriatal injection of 6OHDA. In this model, Doxy given before L-Dopa significantly attenuated LID, and the effect was associated with a reduction in inflammatory markers in the striatum.

According to the concept of a multifactorial approach to neurodegenerative disorders, Doxy combines two different pharmacological activities in a single molecule: the inhibition of α-syn aggregation and the control of inflammation [273]. The drug is ready for a clinical challenge [256]. Its safety profile, confirmed by recent chronic treatment [274,275] and favorable pharmacokinetics [276] make Doxy an excellent candidate for PD subjects in the early phase or even in the prodromal phase.

Other compounds also have the multi-target activities of Doxy, including the anti-oligomeric activity against α-Syn0s [277,278]. Rifampicin is a well-known antibiotic with interesting pharmacological activities independently from the original anti-infectivity action. Rifampicin interacts with oligomers of different origins, including α-Syn0s, as demonstrated in vitro and in vivo [279,280]. This was associated with anti-inflammatory activities in PD and LDB models [281,282,283,284]. The anti-inflammatory activity of rifampicin in a neurological context has been shown by Bi et al. (2011) [285] using microglia BV2 cells alone or in combination with neurons to examine the neuroprotective consequences against the microglial reaction.

More recently, in zebrafish injected with rotenone, the attenuation of mitochondrial dysfunction by rifampicin was associated with a reduction in inflammatory reactivity. A similar effect has been described in human microglial cells stimulated by rotenone: in this case, the control of inflammation depended on activation of the V-type proton ATPase subunit isoform 1 (Atp6v0a1,) a subunit involved in lysosomal mechanisms [268]. In LPS-induced cognitive and motor impairment in mice, the protective effect of rifampicin appears to be mediated by the TLR4/MyD88/NF-κB signaling pathway. In synucleinopathies and the other neurodegenerative disorders, despite abundant experimental results, the clinical efficacy of rifampicin has been investigated little, with contradictory results [17,286,287,288].

## 5. Conclusions

The central role of α-Syn in PD and associated disorders is well established; genetic and neuropathological findings indicate that α-Syn aggregation is an initial causal event in the early phase of the disease. Alpha-Syn aggregation follows the nucleation (or seeding) model from monomer through oligomers to insoluble fibrils, common to other protein-misfolding diseases. The intracellular compartmentalization of the LB does not hamper the spread of LB pathologies through the brain and from outside (ENS) to the CNS. Both propagation from the periphery to the CNS and in the opposite direction are compatible with the development of PD, and numerous experimental studies, though not all, have proved that fibrillar α-Syn or PD brain-purified material, injected in the ENS or CNS, can spread in both directions. Numerous experimental findings point to α-SynOs, the soluble form of α-Syn aggregate, as the main agent of the pathological spread and cell-to-cell transmission, although exosomal transport can also involve other forms of aggregates [289]. 

Alpha-SynOs have the direct neurotoxic ability to induce neuronal dysfunction, with mechanisms shared by the other pathological oligomers [17,126]. Several findings indicate that cell-to-cell spread, intracellular seeding and neurotoxicity are distinct mechanisms. Neuropathological examinations of PD brain show clear differences between the distribution of α-SynOs and LB pathologies [73]. Together with α-Syn deposits in the early phases of synucleinopathies, an excessive inflammatory reaction sustains the pathology. Our preclinical models show that general inflammation can enhance the α-SynOs’ cerebral toxicity and that inflammation promotes the propagation of [290]. Alpha-Syn aggregation and propagation, together with the control of inflammation, are potential targets for disease-modifying treatments in PD and related disorders, which so far have been equipped only with symptomatic tools.

As generally happens with neurodegenerative diseases, numerous approaches to treat PD or MSA, including anti-amyloidogenic, small molecules or immunotherapy, have given promising results in preclinical studies but have not translated successfully to human pathological conditions. Multiple reasons explain these failures, starting from the disease complexity, the timing of treatment, selection of patients and identification of the appropriate therapeutic target and tools. All of these aspects and others in the treatment of synucleinopathies can be refined [248]. Specific efforts should focus on treatment timing: the formation of α-syn deposits and inflammation are both early events in the disease, and intervention in a prodromal phase is recommended. Another fundamental consideration involves multifunctional treatments combining diverse pharmacological targets in a single molecule or using separate compounds. Doxy is a good example to test this strategy; solid findings support its dual anti-inflammatory and anti-amyloidogenic functions with the advantages of repurposing a drug already well-known in clinical practice.

Growing knowledge about the biology of α-Syn and inflammation in neuropathological conditions continuously offers opportunities to develop new therapeutic strategies and earlier diagnostic tools [291], though they all need to be tested faster at the clinical level. The scientific community, regulatory systems and drug companies are invited to share knowledge and skills to improve the clinical protocols and speed up the evaluation of candidate treatments to interfere with the pathological causes of synucleinopathies and the other neurodegenerative disorders.

## Figures and Tables

**Figure 1 ijms-24-05914-f001:**
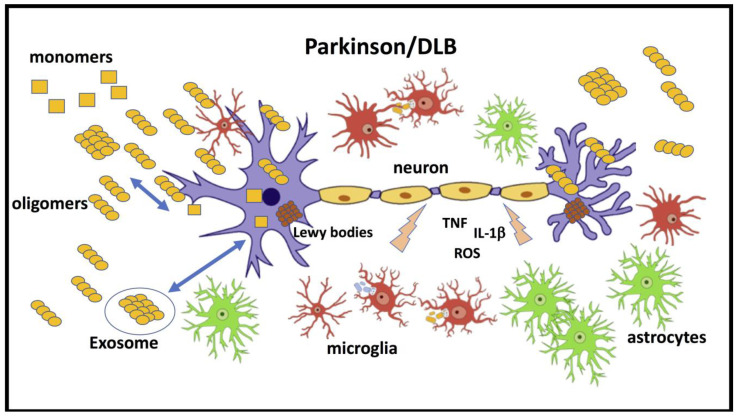
The pathological scenario in synucleinopathies. α-Syn in various forms, monomeric, oligomeric and fibrils, interact with neurons and glial cells. Neuronal dysfunction is induced by direct α-Syn oligomers, originating from outside or inside the cells. The permanent activation of microglial cells can induce the production of factors with detrimental effects on neuronal function, and astrocyte activation or dysfunction may contribute to neuronal damage.

**Figure 2 ijms-24-05914-f002:**
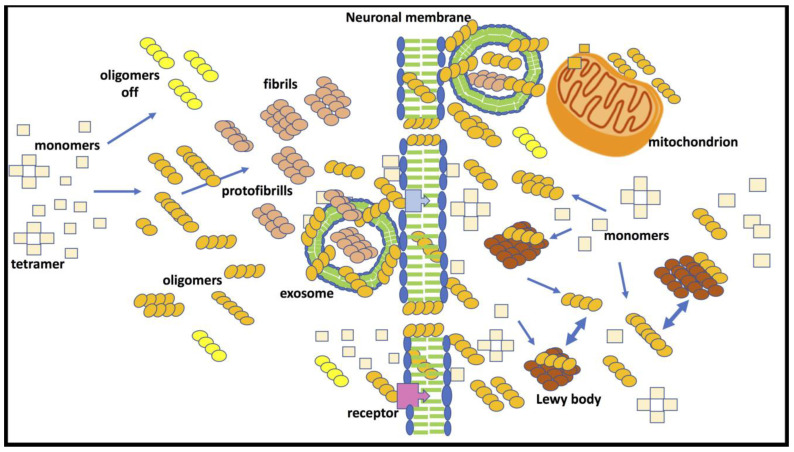
α-Syn and neuronal dysfunction. The fibrillogenic process from monomers to Lewy bodies passes through soluble aggregates with α-Syn oligomer as an intermediate conformer. These soluble aggregates may interact directly with neuronal membrane through specific acceptor, pore formation or unspecific interference with double lipid structure. Endocytosis and exocytosis through exosomes may contribute to the formation of Lewy bodies and cell-to-cell spread of the pathology. The progress of α-Syn oligomers to protofibrils and fibrils is not inevitable, and in specific conditions, stable oligomers can be formed (oligomer off) with no further assembling.

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
