# Peer review of "Alpha Synuclein: Neurodegeneration and Inflammation"

_ijms, 2023, doi:10.3390/ijms24065914_

Round 1
Reviewer 1 Report
In this paper, Gianluigi have described the role of alpha synuclein in neurodegeneration and inflammation. It is interesting, but there are some following problems.
1. In the CNS, the olfactory bulb may have the earliest pathological changes. What is the distribution of α-Syn pathology/α-SynO in olfactory bulb? Previous studies have shown that α-Syn pathology can spread from the olfactory bulb to other brain regions, and how aboutα-Syn pathology/α-SynO 's spread from the olfactory bulb to other brain regions?
2. What are the specific mechanisms of α-SynO neurotoxicity? Can it become a potential target for treatment?
3. α-Syn transmission and deposition was constantly associated with inflammation induced by α-Syn, and in a vicious circle inflammation promoted α-Syn pathology propagation. According to the author's analysis, is there any direct research support that intestinal inflammation promotes the transmission of α-SynO from peripheral to CNS?
4. What is ICV? What are the advantages of this method?
5. What is the relationship between peripheralα-SynO and CNSα-SynO? How to choose this target during treatment?
6. There are some mistakes in the paper.
In various experimental conditions in has been reported
α-SynOs induced cognitive decline seen in the by object recognition test (ORT)
The stimulation of TLRs is part of the activation of the NOD-like receptor pyrin domain containing 3 (NLRP3) inflammasome. (Ravichandran and Heneka, 2021).
The different mechanisms involvingα-SynOs influenced by TLR-2 include: microglia and astrocytic activation; neuronal autophagy and α-Syn transmission from neuron to neuron or from neurons to glial cells (Kwon et al 2019.
Author Response
In this paper, Gianluigi have described the role of alpha synuclein in neurodegeneration and inflammation. It is interesting, but there are some following problems.
In the CNS, the olfactory bulb may have the earliest pathological changes. What is the distribution of α-Syn pathology/α-SynO in olfactory bulb? Previous studies have shown that α-Syn pathology can spread from the olfactory bulb to other brain regions, and how aboutα-Syn pathology/α-SynO 's spread from the olfactory bulb to other brain regions?
The role of olfactory bulb in spreading has been highlighted (p.9)
What are the specific mechanisms of α-SynO neurotoxicity? Can it become a potential target for treatment?
The neurotoxicity of α-SynO has been further illustrated (p.13)
α-Syn transmission and deposition was constantly associated with inflammation induced by α-Syn, and in a vicious circle inflammation promoted α-Syn pathology propagation. According to the author's analysis, is there any direct research support that intestinal inflammation promotes the transmission of α-SynO from peripheral to CNS?
Several evidence have shown that peripheral inflammation can influence PD progression, in a paper now cited intestinal inflammation influenced cerebral α-Syn pathology, however a direct evaluation of the effect on α-SynO is not yet been investigated
What is ICV? What are the advantages of this method?
The intracerebroventricular administration is necessary when the treatment by peripheral injection can not reach the brain or when specific intracerebral effect is investigated.
What is the relationship between peripheralα-SynO and CNSα-SynO? How to choose this target during treatment?
For the several aspects analyzed in the review at moment it is not reasonable to distinguish a treatment anti-oligomers central or peripherally.
In the review we hypothesize that α-SynO can be responsible of the α-Syn pathology spreading within and outside the brain, as illustrated this is compatible with PD pathogenesis
There are some mistakes in the paper.
In various experimental conditions in has been reported
α-SynOs induced cognitive decline seen in the by object recognition test (ORT)
The stimulation of TLRs is part of the activation of the NOD-like receptor pyrin domain containing 3 (NLRP3) inflammasome. (Ravichandran and Heneka, 2021).
The different mechanisms involvingα-SynOs influenced by TLR-2 include: microglia and astrocytic activation; neuronal autophagy and aα-Syn transmission from neuron to neuron or from neurons to glial cells (Kwon et al 2019).
Correct
Reviewer 2 Report
1) Author provides a concise insight to recent advancements under conceptual framework of the adopted term ‘oligomeropathies’; however, it is suggested and preferred if author could include a tabulated form of the discussed information on size of the monomers vs oligomers and their relative tendencies and features in context to their pathological relevancy.
2) Lines 84-86; Role of alpha-Syn monomers or oligomers, and their respective interactions with immunological system needs more specific citations, while simultaneously their roles in PD-pathology are not specified very clearly.
3) Fig.1. the highlighted role of activated microglia and it’s so far known attributed role or perspectives needs more specification.
4) Author mentions about the possibility of pathology spread mediated via exosomes, does pathological-meric forms of the protein transmitted usually found clogged to the vesicle membrane or as vesicle content, please specify in the manuscript and cite the corresponding compelling evidences. Also, please adjust the Fig.2. accordingly.
5) Do all the post translational modifications have similar level of virulence or tendency of oligomerization in different PD models, please update the manuscript accordingly.
6) Lines 210-12, does 200nm fragments have preferential affinity to a special class of lipids on the membrane, if so, please highlight in the manuscript accordingly.
7) Lines 626-29, Doxy’s effect on Ab-plaques is not really required for this review, moreover there are studies available showing irrelevancy or loose correlation of plaque formation verses AD pathophysiology, therefore author may consider removing these comments.
8) Please revisit the manuscript for issues with figures, grammar and typo etc.
Author Response
Review 1
Author provides a concise insight to recent advancements under conceptual framework of the adopted term ‘oligomeropathies’; however, it is suggested and preferred if author could include a tabulated form of the discussed information on size of the monomers vs oligomers and their relative tendencies and features in context to their pathological relevancy.
According to review suggestion I tried to compose a table with monomers vs oligomers biological activities in the pathological context but as expected almost the totality of results are associated with oligomeric or fibrillar form of a-synuclein. In the new version of the paper, I better clarify this point (p.4)
2) Lines 84-86; Role of alpha-Syn monomers or oligomers, and their respective interactions with immunological system needs more specific citations, while simultaneously their roles in PD-pathology are not specified very clearly.
As suggested I add some specific references at the end of the sentence, the concept has been explained in the paragraph dedicated to inflammation.
3) Fig.1. the highlighted role of activated microglia and it’s so far known attributed role or perspectives needs more specification.
The general role of microglia in neurodegenerative disorders and in relation to a-syn has been explained by the addition of a paragraph at p.16
4) Author mentions about the possibility of pathology spread mediated via exosomes, does pathological-meric forms of the protein transmitted usually found clogged to the vesicle membrane or as vesicle content, please specify in the manuscript and cite the corresponding compelling evidences. Also, please adjust the Fig.2. accordingly.
The aggregated forms of a-syn is associated with exosomes externally or inserted within the membrane, however internal accumulation has been also observed. This information is now added (p.11 ) and Fig.2 modified accordingly.
5) Do all the post translational modifications have similar level of virulence or tendency of oligomerization in different PD models, please update the manuscript accordingly.
The influence of post translational modifications have been now illustrate (p.7-8 )
6) Lines 210-12, does 200nm fragments have preferential affinity to a special class of lipids on the membrane, if so, please highlight in the manuscript accordingly.
In the cited paper (Emin et al 2022) it was simple tested the activity of various -synuclein sucrose purified fractions to induce Ca++ efflux from artificial liposome previous loaded with fluorescent sensible to Ca++and 200nm fragments were the most active showing its capacity to induce membrane pemeabilization.
7) Lines 626-29, Doxy’s effect on Ab-plaques is not really required for this review, moreover there are studies available showing irrelevancy or loose correlation of plaque formation verses AD pathophysiology, therefore author may consider removing these comments.
Done
8) Please revisit the manuscript for issues with figures, grammar and typo etc.
The manuscript has been revised by our English editor
Round 2
Reviewer 1 Report
The author has answered most of the reviewers' questions.
Author Response
According to the review comment, the paper has been revised with attention, the changes are highlighted in the manuscript